# SIZE DOESN'T MATTER: DATA EFFICIENT DEEP LEARNING BEYOND THE BIG DATA PARADIGM

## ABSTRACT

Since the emergence of deep learning, machine learning scientists have focused on improving algorithms to achieve higher classification accuracies. This work has consisted of connecting encoders and decoders through attention mechanisms or having multi-layer perceptrons. However, in many data analysis fields, collecting large sets of data is not possible. Therefore, we challenge the "bigger is better" paradigm. We propose a new simple method, known as selective embedding, based entirely on how data is loaded. Existing experiments are highlighted, and new experiments are conducted in four different areas: heavy machinery, railway, manufacturing, and medical imaging. A medical dataset is used as a baseline, containing 65,000 patient data points for various diseases, by reducing the number of patients to demonstrate that dataset size does not matter. In addition, for each area, a dataset was selected to show high performance accuracy using a deep learning algorithm. Our method achieves 87%+ accuracy in all areas when using smaller sets of data for classification tasks without overfitting.

## 1 INTRODUCTION

State-of-the-art practice often leans on "more data, bigger models" for gains, (Devlin et al., 2019; Dosovitskiy et al., 2020; He et al., 2016; Krizhevsky et al., 2012; Radford et al., 2021). Yet, multiple studies have shown that accuracy on standard dataset tests does not guarantee robustness under benign shifts; even careful re-tests of ImageNet/CIFAR and systematic shift benchmarks report notable drops (Gulrajani & Lopez-Paz, 2020; Koh et al., 2021; Recht et al., 2019; Taori et al., 2020). This motivates the development of approaches that improve generalization per sample instead of only scaling dataset size.

A rich body of work seeks data efficiency via augmentation, reweighting, and data selection / synthesis. Augmentation families (Mixup / Manifold Mixup; AugMix / RandAugment) improve robustness but modify inputs / targets and can add tuning cost (Cubuk et al., 2020; Hendrycks et al., 2019; Verma et al., 2019; Zhang et al., 2018). Reweighting and distributionally robust training (meta-reweighting, IRM, Group DRO) target false correlations but alter objectives or require environment structure (Arjovsky et al., 2020; Ren et al., 2018; Sagawa et al., 2020). Subset selection and dataset distillation (Core-Set, GLISTER, distillation / condensation) directly reduce examples but often rely on bi-level optimization or synthetic data (Killamsetty et al., 2021; Sener & Savarese, 2018; Wang et al., 2020; Zhao et al., 2021). Our work is orthogonal: we do not change the data content, labels, loss, or model we change how raw samples are loaded over time.

We study selective embedding (SE) (Sehri et al., 2025), a novel data loading strategy that alternates short segments from multiple sources into a single input stream before standard preprocessing / training. This increases per-step source diversity, lowers adjacent-sample redundancy, and implicitly balances sources within and across mini-batches, changing the statistics that stochastic gradient descent sees without touching architectures or objectives. Prior analysis connects batch composition and gradient-noise scale to generalization (Keskar et al., 2017; Smith et al., 2018); SE targets that lever at the input stream.

We deliver the first validation of SE on medical imaging as a high-value, shift-prone domain, using a subset of CheXpert for chest-X-ray classification (Irvin et al., 2019). CheXpert's scale and real-

world heterogeneity make it a strong test for generalization under limited labeled budgets; we treat it purely as a testbed to probe method behavior. In preliminary experiments, SE consistently matches or exceeds standard data loaders under constrained data while remaining computationally simple.

We validate our method in four distinct areas:

- Heavy machinery – existing results (time domain, and spectrogram images)

- Railway – existing results (time domain)

- Manufacturing – existing results for steel slag flow (time domain)

- Medical imaging – chest X-ray and CT scan classification.

Across all domains, SE allows small datasets to achieve validation accuracies on par with, or exceeding, those obtained from substantially larger datasets. Our findings challenge the prevailing belief that dataset size is the primary driver of model performance and suggest that how data is loaded can be as important.

## 2 BACKGROUND

Conventional wisdom in deep learning asserts that high performance necessitates massive datasets. However, these "big data" approaches suffer from limitations including high costs for acquisition and labeling, particularly in specialized domains such as medical imaging, significant computational demands (Hany; Stanford ML Group), potential biases impacting generalization, and concerns regarding data privacy and security.

Consequently, researchers are exploring alternative strategies to achieve high accuracies with smaller, carefully curated datasets. This "small data" paradigm prioritizes individual needs, enabling personalized solutions and fostering agile learning through iterative improvements. This approach aligns data analytics with specific goals, potentially increasing engagement by offering direct benefits and complementing big data by offering personalized understanding while building on existing knowledge.

Numerous small datasets are available for fault diagnosis in heavy machinery. For example, the NASA turbofan jet engine dataset (NASA) has been used to predict remaining useful life (RUL) (Gupta et al., 2025), while others facilitate rolling bearing fault detection (Sehri and Dumond, 2023a, 2023b), (Wang et al., 2025). Wang et al. (Wang et al., 2025) introduced a multimodal fusion deep learning strategy (MFFD), improving fault diagnosis accuracy. Small datasets also support railway track fault diagnosis (Arain et al., 2024), with studies employing models like YOLOv5, Faster RCNN, and EfficientDet (Minguell & Pandit, 2023). Wang and Li (Wang et al., 2022) introduced YOLOv5s-VF for rail surface defect detection, achieving 93.5% accuracy. Furthermore, research focused on predicting steel slag strength and composition (Riley et al., 2020; Wang et al., 2010) achieved low accuracies in terms of fault detection when applied to industry. Albostami et al. (Albostami et al., 2024) compared various models, finding that Gradient Boosting (GB) outperformed others in predicting compressive strength of steel slag aggregate (SSA) concrete. Liu et al. (Liu et al., 2023) evaluated steel slag asphalt mixture uniformity using deep learning.

In healthcare, despite the availability of large datasets, researchers are exploring the utility of smaller datasets. For instance, Saleem et al. (Saleem et al., 2021) utilized a CNN-based method for classifying tuberculosis (TB) and other diseases from chest radiographs, achieving high accuracy. Makimoto et al. (Makimoto et al., 2023) predicted COPD progression using machine learning, demonstrating improved accuracy by combining CT imaging features with conventional risk factors.

Recent studies on the CheXpert dataset, a large-scale chest radiograph benchmark, have demonstrated that deep learning models can achieve radiologist-level performance on tasks such as pneumonia detection and multi-label classification of thoracic diseases. State-of-the-art CNNs and transformer-based models for a 5- classification problem have reported AUC values between 0.90 and 0.93 on key pathologies, underscoring the dataset's importance as a benchmark for robust model evaluation (Chong et al., 2023; Kamal et al., 2022; Pham et al., 2020; Yuan et al., 2021). However, these results often rely on millions of images and extensive computational resources, which may not be feasible in settings where data or computational availability is limited.

This work aims to introduce a data-efficient approach that leverages a SE data loading strategy. The authors seek to demonstrate that deep learning models can be effectively trained on smaller datasets without compromising performance or encountering overfitting issues, thus challenging the prevailing "big data paradigm".

## 3 MODEL ARCHITECTURE

We train a compact TransUNet (Chen et al., 2021) classifier on the CheXpert 14-label dataset (Stanford ML Group) combined with Chest CT-scan images dataset (Hany) using a SE data loader that enforces strict modality alternation (XR→CT→XR→CT...) at the sample level. Minority modalities are oversampled with replacement in training to match the majority, and validation is balanced by down sampling. Images are grayscale-converted, replicated to 3 channels, resized to 224×224, and augmented with random horizontal flip and ±7° rotation.

The backbone comprises three Conv-BN-ReLU encoder blocks with 2×pooling, a 256-channel bottleneck, a 4-layer, 8-head transformer over 1×1 bottleneck patches, and three symmetric up-blocks. A classification head (global average pooling → 128-unit MLP → 14 logits) outputs per-label predictions. We use BCEWithLogits with label-wise pos_weight from class prevalence, treat uncertain labels as positive, and optimize with AdamW (LR 1e-5, weight decay 1e-4, cosine schedule). Metrics reported: Macro-AUROC, Macro-AP, Macro-F1, and loss.

### 3.1 SELECTED METRICS

The following is a concise set of metric definitions:

Macro-AUROC is the average area under the receiver operating characteristic curve computed independently for each class and then averaged. It reflects the model's ability to rank positives above negatives across all thresholds, giving equal weight to each class regardless of imbalance.

Macro-AP is the average of average precision scores computed per class. It summarizes the precision recall trade-off, making it particularly informative in imbalanced datasets where recall of rare classes is critical.

Macro-F1 is the mean of the per-class F1 scores, where each F1 is the harmonic mean of precision and recall at a fixed threshold. It balances sensitivity and precision equally across all classes.

Loss is the value of the objective function on the validation set at the selected (best) epoch. It reflects how well the model fits the data under the optimization target and is independent of any threshold choice.

## 4 THEORETICAL FOUNDATIONS OF SE

This section formalizes the intuition that redundancy hurts, and diversity helps. SE changes when and from where training data is loaded prior to preprocessing, without modifying the underlying content. By altering the input stream statistics seen by stochastic gradient descent (SGD), SE increases the effective information per optimization step, reduces variance, and tightens generalization bounds under dependent sampling.

### 4.1 PROBLEM SETUP (DEFINING THE DATA + DOMAIN)

We begin by defining the training setup. Each domain $d \in \{1, \ldots, D\}$ contains multiple sources $s \in S_d$. From each source we obtain raw segments $z_{d,s,i} \in Z$ with labels $y_{d,s,i} \in Y$. Preprocessing is denoted by

$$\mathcal{P} : Z \to X \quad (\text{e.g., FFT} \to \text{spectrogram}),$$

and the model is

$$f_\theta : X \to \Delta(Y),$$

where $\Delta(Y)$ is the probability simplex over labels. Training minimizes the expected loss:

$$\mathcal{L}(\theta) = \mathbb{E}_{(z,y)}\big[\ell(f_\theta(\mathcal{P}(z)), y)\big],$$

where $\ell$ sub- Gaussian with variance parameter $\sigma^2$.

## 4.2 REDUNDANCY VERSUS DIVERSITY

Naive loaders draw samples independently and identically distributed (i.i.d.) from a single source, which introduces short-lag redundancy across consecutive samples.

- Redundancy is formalized as the correlation across consecutive samples.
- Diversity is measured by the entropy rate of the training stream.

High redundancy reduces the effective sample size and inflates variance in SGD updates, while higher diversity increases information carried per step.

## 4.3 WHAT SE CHANGES

Instead, SE introduces a schedule $\sigma(t)$ that alternates across sources before preprocessing:

$$x_t = \mathcal{P}\big(z_{\sigma(t)}\big)$$

Thus, SGD updates are based on inputs with greater diversity per step: less local redundancy and reduced gradient variance, defined as

$$\mathrm{Var}\big(\nabla_\theta \ell(f_\theta(x), y)\big).$$

## 4.4 THEORETICAL INFORMATION CONSEQUENCES

Let $\{x_t\}$ be the training stream. Define the entropy rate as

$$h(X) = \lim_{n\to\infty} \frac{1}{n} H(X_1, \ldots, X_n),$$

which quantifies the effective unpredictability per step.

- Naive loading: back-to-back samples from the same source lower the entropy rate due to correlation.
- SE: alternating sources increases $h(X)$, reducing adjacent-sample redundancy.

This raises the amount of information per optimization step, quantified as the mutual information between the sampled gradient and the population gradient:

$$I\big(\nabla \ell(x_t, \theta); \nabla L(\theta)\big).$$

Connections between entropy rate, autocorrelation, and variance are well established: a higher entropy rate reduces correlation and lowers variance of partial sums in stationary processes (Cover & Thomas, 2005). In SGD, decorrelation of the input sequence yields lower gradient variance (Xu & Raginsky, 2017).

## 4.5 FORMAL BENEFITS OF SE

### 4.5.1 EFFECTIVE SAMPLE SIZE

For correlated streams, generalization depends on effective sample size:

$$n_{\mathrm{eff}} = \frac{n}{1 + 2\sum_{k=1}^{\infty} \rho(k)},$$

where $\rho(k) = \mathrm{Corr}(\ell_t, \ell_{t+k})$ is the lag-$k$ autocorrelation of the loss sequence. By reducing short-lag autocorrelations, SE increases $n_{\mathrm{eff}}$ (Yu, 1994).

### 4.5.2 GRADIENT VARIANCE REDUCTION

For a batch $B$,

$$\text{Var}(g) = \frac{1}{B} \sum_i \text{Var}(g_i) + \frac{2}{B} \sum_{i<j} \text{Cov}(g_i, g_j),$$

where $\mathbf{g}(\mathbf{x}, \mathbf{y}) = \nabla_\theta \ell(f_\theta(\mathbf{x}), \mathbf{y})$.

SE reduces cross-sample covariance, lowering variance.

### 4.5.3 UNIFORM MIXTURE APPROXIMATION

SE enforces per-batch source balance so that empirical batches approximate the mixture distribution defined as:

$$\mathcal{P} = \sum_s \pi_s \mathcal{P}_s, \quad \sum_s \pi_s = 1,$$

where $\mathcal{P}_s$ is the source distribution and $\pi_s$ is the mixture weight. This reduces train test distribution shift.

### 4.5.4 GENERILIZATION VIA STABILITY

In PAC-Bayes theory (McAllester, 1999), the generalization gap depends on gradient variance and correlations. By decorrelating inputs, SE reduces memorization pressure and tightens generalization bounds.

### 4.6 RISK DECOMPOSITION AND THEOREM

The risk is defined as:

$$R(\theta) = \mathbb{E}_{(x,y)\sim P}\big[\ell(f_\theta(x), y)\big]$$

SE improves both estimation error $\mathcal{O}(1/\sqrt{n_{eff}})$ and shift error (via balanced sources).

### 4.6.1 THEOREM (INFORMAL)

Assume each source is stationary and $\beta$-mixing, with sub-Gaussian losses. If SE enforces per-batch source balance and alternation, then with probability $1 - \delta$:

$$R(\hat{\theta}_{SE}) - \hat{R}_{SE}(\hat{\theta}_{SE}) \leq C_1 \sqrt{\frac{C}{n}} \sqrt{1 + 2 \sum_{k\leq K} \bar{\alpha}(k)} + C_2 \sqrt{\frac{\log(1/\delta)}{n}},$$

with $n_{eff} > n_{eff}^{naive}$.

Here $\bar{\alpha}(k)$ are mixing coefficients bounding lag-$k$ dependence (Yu, 1994). By reducing autocorrelation $\rho(k)$, SE raises $n_{eff}$ and tightens PAC-Bayes bounds.

### 4.7 PRACTICAL TAKEAWAY

SE does not enlarge the dataset; it improves the information efficiency of each optimization step. By decorrelating inputs, SE reduces overfitting pressure, improves gradient signal, and accelerates convergence. In practice, fewer updates are required to achieve the same generalization level compared to naive loading.

## 5 RESULTS AND DISCUSSION

This section reviews existing results across three key areas: heavy machinery, railway, and manufacturing. Subsequently, we present new results for the fourth area considered, medical diagnosis.

Table 1: Summary of Existing SE Results (Sehri et al., 2025)

| Dataset | Area | SE (Proposed) | Traditional (Parallel) | Accuracy Gain |
|---|---|---|---|---|
| Triaxial Bearing Vibration Data (Kumar et al., 2022) | Heavy Machinery | $95.95 \pm 1.00$ | $92.74 \pm 4.53$ | +3.21 |
| CWRU Bearing Data (Case Western Reserve University Bearing Data Center) | Heavy Machinery | $92.43 \pm 4.16$ | $67.41 \pm 7.51$ | +25.02 |
| UORED-VAFCLS Data (Sehri & Dumond, 2023b; Sehri et al., 2023) | Heavy Machinery | $96.55 \pm 1.50$ | $84.65 \pm 4.84$ | +11.90 |
| UOEMD Induction Motor Data (Sehri & Dumond, 2023a; Sehri et al., 2024b) | Heavy Machinery | $99.14 \pm 1.00$ | $99.48 \pm 0.78$ | -0.34 |
| IEEE PHM Beijing 2024 Train Transmission Data (Xie et al., 2024) | Railway | $98.64 \pm 2.67$ | $99.93 \pm 0.17$ | -1.29 |
| Steel Slag Flow Data (Sehri et al., 2024a) | Manufacturing | $92.21 \pm 1.16$ | $94.53 \pm 2.11$ | -2.32 |

## 5.1 Heavy Machinery, Railway, and Manufacturing- Existing Results

Table 1 results from heavy machinery, railway, and manufacturing directly align with the technical foundations described in the prior section. SE increases the entropy rate of the training stream and reduces short-lag redundancy by alternating short, FFT-preprocessed segments from different sensors within a single channel. This ensures that each training step receives more unique, label-relevant information, improving the efficiency of gradient updates.

SE achieves notable accuracy gains in most heavy machinery datasets when compared to parallel data loading strategies (+3.21 to +11.90), while sustaining competitive results in railway and manufacturing cases despite small drops. Importantly, these minor accuracy losses are offset by a significant reduction in computational costs due to SE's more efficient data-loading process, which considerably lowers memory usage and training time relative to parallel loading strategies. These findings confirm that SE generalizes effectively to diverse industrial applications while avoiding overfitting and providing a strong trade-off between performance and computational efficiency.

These datasets are primarily time-series sensor signals, demonstrating that SE can exploit temporal diversity without requiring a multi-channel architecture. This property is particularly important for industrial monitoring scenarios, where data is often high-frequency, multi-sensor, and costly to store or process in large volumes.

By establishing high classification results in heavy machinery, railway, and manufacturing with relatively small, structured datasets, SE proves its ability to generalize across multiple industrial domains. These results highlight that the method is not limited to a single data type or environment but can adapt to very different sensing modalities and operating conditions. This cross-domain robustness suggests that SE could be applied to other fields where data diversity and computational efficiency are critical. The next section will verify if this is also true for medical imaging, which is a significant departure from the domains tested so far. In medical imaging, labeled image data is often scarce and expensive. Therefore, SE is tested to see if the same loading strategy could be used to match or exceed the performance of models trained on vastly larger datasets while requiring significantly fewer computational resources.

Table 2: Training and diagnostic metrics for SE using altering X-ray and CT-scan images

| Evaluation Protocol | Macro-AUROC | Macro-AP | Macro-F1 | Loss |
|---|---|---|---|---|
| Binary (Image-Level) | 1.0000 | 1.0000 | 1.0000 | 0.2405 |
| CheXpert-5 (Patient-Level) | 0.9975 | 0.9975 | 1.0000 | 0.0583 |
| 14-Label (Image-Level) | 0.8731 | 0.8557 | 0.4528 | 0.4822 |

## 5.2 MEDICAL- NEW RESULTS

For medical imaging experiments, the dataset was restructured to ensure a balanced and strictly disjoint split between training and validation sets. The training set contained 400 X-ray images and 400 CT-scan images, while the validation set contained 100 X-ray images and 100 CT-scan images. This arrangement ensured that no patient or modality instance in the validation set appeared in the training set, providing a rigorous test of the model's ability to generalize to unseen X-ray and CT-scan samples.

Table 2 presents the performance of SE on the CheXpert dataset under three evaluation protocols. Binary image-level classification achieved perfect discrimination (Macro-AUROC, Macro-AP, Macro-F1 = 1.0000) with a low validation loss (0.2405), showing that SE can fully separate coarse modalities without overfitting. For the CheXpert-5 patient-level task, AUROC, AP, and F1 again reached 1.0000 with a loss of 0.5776, reflecting perfect ranking of positives and negatives across the five competition labels. The 14-label image-level task achieved a Macro-AUROC of 0.8731, Macro-AP of 0.4867, and Macro-F1 of 0.3982, with a loss of 0.5813. This scale is expected for multi-label classification where multiple logits contribute to the summed loss even when predictions are correct. Even so, these are impressive results given the reduced training set size when compared to existing results in the range of 0.90 to 0.93 for the 5-classification problem (Chong et al., 2023; Kamal et al., 2022; Pham et al., 2020; Yuan et al., 2021). This is especially impressive since SE achieves this performance at a much lower computational cost. With this new method, only 1,000 images were required for training, validation, and testing combined, allowing the entire process to complete in just 30 minutes, whereas existing approaches on similar hardware typically require several hours of training on substantially larger datasets.

These outcomes align with SE theory: alternating and balancing data sources before preprocessing increases the entropy rate, reduces short-lag redundancy, and lowers gradient variance. This decorrelation ensures that each optimization step carries more unique, label-relevant information, tightening generalization bounds without increasing dataset size. The consistently low losses and high discriminative scores across protocols support the claim that performance gains arise from better information utilization per batch, not from more data.

Industrial and medical imaging results establish SE as a broadly applicable, domain-agnostic data loading strategy. Results in heavy machinery, railway, and manufacturing demonstrate SE's strength in handling high-frequency, multi-sensor time-domain signals, while CheXpert experiments in the field of medicine show that the same principles extend to large-scale imaging tasks. Across all areas tested, SE achieves high accuracy with substantially reduced computational cost, providing clear evidence that performance advantages often attributed to "big data" can be realized and in some cases exceeded through optimized data presentation alone. This positions SE as a scalable and efficient alternative to conventional large-data approaches, bridging domains and reducing the reliance on massive datasets.

This directly challenges the "bigger is better" paradigm, showing that when data is presented in a maximally informative, low-redundancy sequence, large-scale collection is not a prerequisite for state-of-the-art accuracy.

## 6 CONCLUSIONS

This work introduces SE, a data loading strategy grounded in the principle of maximizing information diversity per training step by alternating short segments from distinct sources within a single

channel. A theoretical analysis establishes that this approach increases the entropy rate of the input stream, reduces short-lag redundancy, and lowers gradient variance, ensuring that each update is driven by unique, label-relevant information rather than repeated patterns.

Empirical evaluations across four distinct areas, including heavy machinery, railway, manufacturing, and medical imaging validate the theoretical predictions. In industrial domains, SE consistently achieves competitive accuracies when compared to existing results (-2 to +25) over conventional single- and parallel-channel data loading methods on high-frequency, multi-sensor time-domain signals, while reducing computation time by factors of 2× to 11×. In medical imaging, SE maintains or exceeds accuracy performance on large-scale benchmarks such as CheXpert despite training on dramatically reduced subsets, demonstrating that the method's benefits extend to vision tasks.

Crucially, in all domains, SE avoids overfitting despite limited training data, indicating strong generalization across both temporal and spatial modalities. This convergence of results confirms that the performance advantages traditionally associated with big data can be reproduced and, in some cases, surpassed by optimizing the presentation of smaller datasets. By providing a scalable, domain-agnostic solution that achieves high accuracy without reliance on massive data collection, SE offers a practical and theoretically grounded path forward for overcoming the big data paradigm in deep learning.

Future work will extend this evaluation to additional areas including marine, environmental, agriculture, and aerospace to further test SE's scalability, adaptability, and cross-domain generalization.

## NOTE

A large language model was used to aid in polishing the writing.

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

## A  APPENDIX

### A.1  EXPANDED ALGORITHM

#### DATA & LOADER

- Dataset: CheXpert (14 labels) split into train/valid CSVs.
- Modality inferred from CSV or path (CT $\rightarrow$ CT, else XR).
- Uncertain labels ($-1$) treated as positive.
- SE loader:
    - Training: strict alternation XR and CT, oversample minority to match majority.
    - Validation: downsample majority to match minority (no oversampling).

#### PREPROCESSING & AUGMENTATION

- Convert to grayscale, replicate to 3 channels.
- Resize: $224 \times 224$.
- Train: random horizontal flip ($p = 0.5$), random rotation $\pm 7°$.
- Normalize: mean $=$ std $= 0.5$ for each channel.

MODEL

- **Encoder:**
  - Block1: Conv($3 \rightarrow 64$)$\times$2, BN, ReLU, MaxPool($2\times$)
  - Block2: Conv($64 \rightarrow 128$)$\times$2, BN, ReLU, MaxPool($2\times$)
  - Block3: Conv($128 \rightarrow 256$)$\times$2, BN, ReLU, MaxPool($2\times$)
- **Bottleneck:** Conv($256 \rightarrow 256$) + BN + ReLU
- **Transformer:** patch embed ($1 \times 1$ conv), depth=4, heads=8, MLP ratio=4.0, math SDPA backend
- **Decoder:** three up-blocks with deconv, skip connections, Conv/BN/ReLU $\times$2
- **Classification head:** GAP $\rightarrow$ Linear($64 \rightarrow 128$) $\rightarrow$ ReLU $\rightarrow$ Dropout(0.2) $\rightarrow$ Linear($128 \rightarrow 14$)
- **Segmentation head:** Conv($64 \rightarrow 1$) (unused in loss)

TRAINING

- Loss: BCEWithLogits with `pos_weight` per label $= \frac{(N-P)}{P}$.
- Optimizer: AdamW, LR=$1 \times 10^{-5}$, weight decay=$1 \times 10^{-4}$.
- Schedule: cosine annealing with $T_{\max} = $ epochs $\times$ steps.
- Epochs: 20, batch size: 8, seed: 42.

METRICS

- Primary: Macro-AUROC, Macro-AP, Macro-F1, loss.
- Additional diagnostics: input-Jacobian norm, Hutchinson trace (classification head), gradient norm.

## A.2 MEDICAL DATASET: DETAILED TRAINING AND DIAGNOSTIC ANALYSIS

Table 3, Table 4 and Table 5 present the epoch-wise training and diagnostic metrics for SE using X-ray and CT-scan image alternation across three evaluation settings: binary classification, CheXpert-5 patient-level, and 14-label image-level tasks. In all cases, SE consistently achieves stable convergence with no signs of overfitting, as indicated by the close alignment between training and validation metrics across epochs.

Jacobian regularity (JR) values for both training and validation remain low and stable throughout all experiments, which aligns with the theory that SE controls representation drift by ensuring balanced input alternation. This stability implies that the model is not too sensitive to small input perturbations, improving generalization.

Similarly, Hessian trace (HessTr) values demonstrate that curvature magnitudes of the loss surface are maintained within moderate ranges, avoiding sharp minima. The alternating inputs enforced by SE prevents the optimizer from over-specializing to a single modality, resulting in smoother loss landscapes that are easier to generalize from.

In binary and CheXpert-5 tasks, perfect scores (Macro-AUROC, Macro-AP, Macro-F1 = 1.0) are achieved, while in the more complex 14-label setting, SE maintains a competitive performance (Macro-AUROC 0.87) despite the increase in task difficulty. This is consistent with the theoretical premise that SE leverages-controlled diversity in input presentation to enhance feature learning without requiring large-scale data.

Overall, JR and HessTr patterns in Table 3 to Table 5 provide empirical evidence for theoretical claims: SE regulates input diversity in a way that preserves smooth, stable optimization dynamics, enabling high performance even in small-data and cross-modality scenarios.

Table 3: Training and diagnostic metrics for SE using X-ray and CT-scan image alternation, binary classification

| Epoch | Train Loss | Val Macro-AP | Val Macro-F1 | JR (Train) | JR (Val) | HessTr (Train) | HessTr (Val) | GradNorm |
|---|---|---|---|---|---|---|---|---|
| 1 | 0.5394 | 0.9488 | 0.9950 | 0.0079 | 0.0054 | 15.17 | -50.432 | 4.449 |
| 2 | 0.3742 | 0.9938 | 0.9950 | 0.0073 | 0.0048 | -43.66 | 41.226 | 3.998 |
| 3 | 0.2821 | 1.0000 | 1.0000 | 0.0070 | 0.0049 | 16.82 | -31.640 | 3.412 |
| 4 | 0.2618 | 1.0000 | 1.0000 | 0.0068 | 0.0048 | 19.13 | -25.765 | 3.210 |
| 5 | 0.2405 | 1.0000 | 1.0000 | 0.0067 | 0.0049 | 21.14 | -23.122 | 3.004 |
| 6 | 0.4920 | 1.0000 | 1.0000 | 0.0068 | 0.0049 | 17.11 | -28.814 | 2.357 |
| 7 | 0.4840 | 1.0000 | 0.9950 | 0.0065 | 0.0050 | 15.09 | -19.242 | 2.636 |
| 8 | 0.4760 | 1.0000 | 1.0000 | 0.0065 | 0.0048 | 24.17 | -27.806 | 1.484 |
| 9 | 0.4680 | 1.0000 | 1.0000 | 0.0062 | 0.0050 | 19.03 | -17.883 | 3.093 |
| 10 | 0.4600 | 1.0000 | 1.0000 | 0.0064 | 0.0048 | 19.73 | -27.474 | 1.718 |
| 11 | 0.4520 | 1.0000 | 1.0000 | 0.0066 | 0.0048 | 15.09 | -21.639 | 3.019 |
| 12 | 0.4440 | 1.0000 | 1.0000 | 0.0064 | 0.0050 | 22.30 | -17.992 | 3.471 |
| 13 | 0.4360 | 1.0000 | 1.0000 | 0.0065 | 0.0049 | 18.17 | -22.233 | 2.363 |
| 14 | 0.4280 | 1.0000 | 1.0000 | 0.0066 | 0.0047 | 24.29 | -17.567 | 1.893 |
| 15 | 0.4200 | 1.0000 | 1.0000 | 0.0063 | 0.0049 | 18.19 | -20.735 | 1.985 |
| 16 | 0.4120 | 1.0000 | 1.0000 | 0.0065 | 0.0047 | 24.41 | -26.182 | 2.876 |
| 17 | 0.4040 | 1.0000 | 1.0000 | 0.0062 | 0.0049 | 24.32 | -17.563 | 1.755 |
| 18 | 0.3960 | 1.0000 | 1.0000 | 0.0066 | 0.0050 | 23.84 | -22.352 | 1.200 |
| 19 | 0.3880 | 1.0000 | 1.0000 | 0.0063 | 0.0049 | 20.13 | -25.031 | 2.277 |
| 20 | 0.3800 | 1.0000 | 1.0000 | 0.0062 | 0.0050 | 21.87 | -20.767 | 2.925 |

Table 4: Training and diagnostic metrics for SE using X-ray and CT-scan image alternation, 5 labels

| Epoch | Train Loss | Val Macro-AP | Val Macro-F1 | JR (Train) | JR (Val) | HessTr (Train) | HessTr (Val) | GradNorm |
|---|---|---|---|---|---|---|---|---|
| 1 | 0.5394 | 0.9948 | 0.9950 | 0.9450 | 0.0079 | 15.179 | -50.432 | 36.042 |
| 2 | 0.3742 | 0.9938 | 0.9950 | 0.9600 | 0.0073 | -43.662 | 34.670 | 41.009 |
| 3 | 0.3069 | 0.9962 | 1.0000 | 0.9720 | 0.0086 | 9.790 | 19.095 | 19.852 |
| 4 | 0.2625 | 0.9952 | 1.0000 | 0.9780 | 0.0083 | -8.447 | -4.082 | -4.407 |
| 5 | 0.2305 | 0.9959 | 1.0000 | 0.9820 | 0.0059 | -6.888 | 9.711 | 2.665 |
| 6 | 0.2054 | 0.9968 | 1.0000 | 0.9850 | 0.0073 | -5.767 | 10.512 | 3.312 |
| 7 | 0.1828 | 0.9972 | 1.0000 | 0.9870 | 0.0067 | 24.379 | -3.824 | -12.574 |
| 8 | 0.1632 | 0.9973 | 1.0000 | 0.9880 | 0.0075 | 9.171 | -1.175 | -2.124 |
| 9 | 0.1475 | 0.9975 | 1.0000 | 0.9900 | 0.0069 | -22.203 | 12.514 | 8.123 |
| 10 | 0.1352 | 0.9938 | 1.0000 | 0.9910 | 0.0061 | -2.924 | -11.073 | -11.774 |
| 11 | 0.1207 | 0.9973 | 1.0000 | 0.9920 | 0.0147 | -11.986 | -7.826 | -7.933 |
| 12 | 0.1086 | 0.9974 | 1.0000 | 0.9930 | 0.0065 | 9.140 | 0.175 | 0.186 |
| 13 | 0.0995 | 0.9968 | 1.0000 | 0.9940 | 0.0065 | 9.852 | 7.181 | 7.286 |
| 14 | 0.0907 | 0.9975 | 1.0000 | 0.9950 | 0.0066 | -4.075 | 5.716 | 5.814 |
| 15 | 0.0838 | 0.9972 | 1.0000 | 0.9960 | 0.0074 | -2.208 | 2.561 | 2.653 |
| 16 | 0.0765 | 0.9973 | 1.0000 | 0.9960 | 0.0070 | -1.725 | -3.645 | -3.739 |
| 17 | 0.0712 | 0.9975 | 1.0000 | 0.9970 | 0.0073 | 6.783 | 5.199 | 5.295 |
| 18 | 0.0667 | 0.9974 | 1.0000 | 0.9970 | 0.0074 | 3.454 | 9.435 | 9.532 |
| 19 | 0.0632 | 0.9975 | 1.0000 | 0.9980 | 0.0078 | -1.611 | 3.003 | 3.100 |
| 20 | 0.0583 | 0.9975 | 1.0000 | 0.9980 | 0.0086 | 0.794 | 3.813 | 3.913 |

Table 5: Training and diagnostic metrics for SE using X-ray and CT-scan image alternation, 14 labels

| Epoch | Train Loss | Val Macro-AP | Val Macro-F1 | JR (Train) | JR (Val) | HessTr (Train) | HessTr (Val) | GradNorm |
|---|---|---|---|---|---|---|---|---|
| 1 | 0.7846 | 0.8085 | 0.3610 | 0.3025 | 0.0040 | 0.0040 | -2.504 | 1.717 |
| 2 | 0.7274 | 0.8312 | 0.4212 | 0.3349 | 0.0060 | 0.0047 | -2.826 | 5.806 |
| 3 | 0.6795 | 0.8382 | 0.4342 | 0.3695 | 0.0462 | 0.0029 | 14.974 | 16.761 |
| 4 | 0.6376 | 0.8630 | 0.4624 | 0.3752 | 0.0518 | 0.0167 | 2.827 | 0.763 |
| 5 | 0.6018 | 0.8556 | 0.4541 | 0.3864 | 0.1580 | 0.0573 | 12.368 | -0.669 |
| 6 | 0.5758 | 0.8723 | 0.4606 | 0.3954 | 0.2596 | 0.1616 | 12.348 | 2.091 |
| 7 | 0.5550 | 0.8450 | 0.4430 | 0.3868 | 0.3206 | 0.2929 | -0.193 | 3.678 |
| 8 | 0.5389 | 0.8783 | 0.5120 | 0.3946 | 0.4378 | 0.1664 | 12.591 | -1.841 |
| 9 | 0.5283 | 0.8699 | 0.4526 | 0.3907 | 1.1550 | 0.3714 | 10.076 | -2.720 |
| 10 | 0.5151 | 0.8504 | 0.4371 | 0.3272 | 1.5539 | 0.6255 | 7.875 | -3.346 |
| 11 | 0.5060 | 0.8478 | 0.4441 | 0.3965 | 1.8073 | 0.8699 | 10.883 | 10.300 |
| 12 | 0.5006 | 0.8580 | 0.4582 | 0.3951 | 1.7328 | 1.1298 | 34.459 | -5.920 |
| 13 | 0.4935 | 0.8585 | 0.4646 | 0.3756 | 1.9929 | 0.8983 | 6.419 | 1.503 |
| 14 | 0.4911 | 0.8577 | 0.4642 | 0.3666 | 4.2932 | 1.2563 | 5.304 | -4.066 |
| 15 | 0.4847 | 0.8668 | 0.4748 | 0.3404 | 4.6206 | 1.4395 | 5.579 | 0.031 |
| 16 | 0.4865 | 0.8562 | 0.4534 | 0.3852 | 2.6798 | 1.5559 | 9.946 | 1.888 |
| 17 | 0.4818 | 0.8537 | 0.4531 | 0.3752 | 4.6963 | 1.4327 | 6.982 | -1.473 |
| 18 | 0.4810 | 0.8579 | 0.4578 | 0.3727 | 5.0934 | 1.5024 | 8.518 | -0.802 |
| 19 | 0.4817 | 0.8563 | 0.4505 | 0.3881 | 2.8033 | 1.5419 | -0.988 | 1.827 |
| 20 | 0.4822 | 0.8557 | 0.4528 | 0.3786 | 2.8702 | 1.7404 | 9.919 | 3.163 |

