# OpenReview forum: "Size Doesn't Matter: Data Efficient Deep Learning Beyond the Big Data Paradigm"
_ICLR.cc/2026/Conference — ICLR 2026 Conference Withdrawn Submission_

### Official Review · Reviewer_7GxM · 2025-10-19

**Soundness:** 1
**Presentation:** 1
**Contribution:** 1
**Rating:** 0
**Confidence:** 5

**Summary:**

A study that reads as if most of it was LLM generated.
The main idea - if any - seems to be a new way to compose mini-batches.
However, unfortunately the presentation is so obscure that it is difficult to say what the contribution is.

**Strengths:**

The authors admit that they used an LLM in the writing process.

**Weaknesses:**

This is a clear reject. I am not a native speaker myself, so I am sorry to write that communication is also an important part of research.
This is a submission that reads as if it was in large parts LLM generated.

The text is an incoherent concatenation of unclear statements, some of which are wrong.
The mathematical statements lack support, they are not proven. The notation is inconsistent.
Even the main algorithm is not clearly explained.

Just some examples:

- Abstract: „Since the emergence of deep learning, machine learning scientists have focused on improving algorithms to achieve higher classification accuracies.“ No, I would say ML researchers always have been doing this.
- „Conventional wisdom in deep learning asserts that high performance necessitates massive datasets.“ I disagree, in particular if one does not consider generative AI.
- Background: The background gives the impression as if small data sets would be something special. They aren’t. The UCI benchmark repository is full of - by today’s standards - small data sets.
- 4.5.4: No, standard PAC-Bayes analysis does not consider „gradient variance“.
- 4.6.1 Proof?
- ... ... ...

**Questions:**

Where does PAC-Bayesian analysis consider „gradient variance"?
Can you provide a rigorous proof of the statement in 4.6.1?

---

### Official Review · Reviewer_ikD8 · 2025-10-27

**Soundness:** 1
**Presentation:** 3
**Contribution:** 1
**Rating:** 0
**Confidence:** 3

**Summary:**

The paper follows up on an earlier paper named "Selective embedding for deep learning", or SE for the purpose of this review. SE is a proposed 'data loading' strategy that could be added as a wrapper to any model with no change to it. SE alternates short segments of data from multiple sources (and data types or modalities) into a single input stream prior to any processing. The idea decorrelates inputs from either interleaved stream to improve the entropy rate of data by reducing the variance (svd style). The intention is apparently to intended to challenge the "bigger is better" paradigm, in terms of the amount of data. Experiments are diverse and on a smaller number of examples for a comparable   statistical strength.

**Strengths:**

. The method sppears original, and I have not otherwise read literature in the domain generalization theme where such multi-source data  are interleaved.

. The 11x  speedup reported

**Weaknesses:**

. The idea that data are 'mixable' at anything above the byte level is  a stretch to belief. In disregard to data type, their  information density relative to that type, their representational nature (e.g. 3D vs 1D), it does no appear sound to just mix them up.

I would have  understood if data were being projected  to a unified embedding  space before such interleaving were performed. But it is as crude as the concatenation it derides.

I use the term embedding in a more general manner, to stricke a contrast with what the paper's quite non standard usage of the term.

. Other weaknesses include small scale experiments e.g. on a 800/200 example split. The "size" emphasis is weakened. If these are all images, load-time mixing is moot. Here, CT and MRI is being mixed, to the horror of anybody who understands how different these modalities are.
The paper doesn't divulge that in Table 1 the traditional ' parallel' loading went through  an   equivalent pipeline, or at least the architecture changed.

. Experiments aren't extensive or on private data

**Questions:**

. How does the paper differ from 'se for dl' in its core idea? I understand that the writing is due to an LM and therefore a lot improved.

---

### Official Review · Reviewer_Cvbb · 2025-10-28

**Soundness:** 1
**Presentation:** 1
**Contribution:** 1
**Rating:** 0
**Confidence:** 3

**Summary:**

This paper lacks significant novel contributions. The proposed method, Selective Embedding (SE), is not introduced here but originates from prior work (Sehri et al., 2025). The industrial experiments on heavy machinery, railway, and manufacturing domains are directly borrowed from that earlier publication, without new analysis or ablations. The only new experiment is on the CheXpert medical imaging dataset. However, they use an extremely small training set and achieve implausibly perfect validation metrics, raising strong concerns about overfitting and lack of generalization. Hence, the central  “size doesn’t matter” claim is not convincingly supported by any means. Overall, while the paper presents an interesting perspective on data efficiency, it does not offer a clear methodological advance or rigorous empirical validation to justify its main conclusions.

**Strengths:**

Poorly written paper. The clarity is fair, and it’s clearly not delivering any evidence to support their claims.

**Weaknesses:**

The core claim, “size doesn’t matter”, is not supported in any way in this paper. The borrowed experiment results like heavy machine and railroad cases don’t support the size claim, but reveal the order of data loading and variety in data sample impact efficiency. There’s no mention of a subset of the data used in the training, which would have provided some evidence that size can indeed be reduced. So the result doesn’t support the “size doesn’t matter” claim. The only new experiment with the CheXpert dataset was conducted on a smaller subset, but without testing if the result can be generalized to a bigger dataset. The fact that they got AUROC of 1.0 strongly suggests overfitting. Hence, this experiment doesn’t support their claim either. In the paper they did mention that smaller datasets can be used for customization, but that doesn’t justify that larger datasets are not needed, either.

**Questions:**

Please check my reviews on weaknesses.

---

### Official Review · Reviewer_j8jt · 2025-10-30

**Soundness:** 1
**Presentation:** 1
**Contribution:** 1
**Rating:** 2
**Confidence:** 4

**Summary:**

This paper studies how interleaving data from multiple modalities using an existing method called "selective embedding" into short segments helps with learning when compared to using continuous streams of data. Experiments on different datasets seem to show this is helpful. There are theoretical results that discuss some of the features of the SE method.

**Strengths:**

* The motivation to question scale of datasets is interesting.
* SE method is simple enough. It can be useful when used in appropriate tasks.

**Weaknesses:**

* **Unclear hypothesis and motivation:** The authors set out to address two main questions: "data efficient deep learning" and to show how "redundancy hurts, and diversity helps". However, it is unclear how the rest of the paper -- particularly the empirical evidence is proving this? Where are the results for showing data efficiency? The authors just mention "reducing computation time by factors of 2x to 11x". Where are the experiments for this?

* **Double claiming novelty:** The authors say  they propose a new method "selective embedding" which is based on existing work referenced [1]. Are the authors claiming this current work does a variation of SE in [1]? It is very unclear, and ethically problematic to claim novelty over existing work.

* **Theoretical claims are confusing:** I could not understand the point of Sec. 4 with theory that is very confusing. The results in Sec. 4 are disconnected, to say the least. It is also unclear how these theoretical results inform the experiments reported later in the paper. How are these theoretical insights evaluated in the specific experiments performed?

* **Experiments do not support the claims:** Results in Table 1 and 2 are not very informative. It is unclear as to what the datasets are in Table 1. And in Table 2, are the authors comparing with the original Chexpert dataset? Or just the train-val-test using 1000 images. If so, how can they claim that SE helps to achieve similar performance when they are not comparing the same data splits?

* **Missing baselines/ improper experimental design:** There are simple baselines like random sampling or other coreset methods that have been studied extensively to improve data efficiency. There are no other baseline methods reported. I would be curious to see how SE fares compared to simple methods.

### References

[1] Mert Sehri, Z. Hua, Felipe de A. Boldt, and Patrick Dumond. Selective embedding for deep learning. 2025.

## Other comments

* What is the point of Sec 4.6.1? How are these assumptions relevant here? What is $\beta$ mixing with sub-Gaussian loss in this problem setting?
* Dataset is not clearly explained. What are the "heavy machinery". "Railway", and "manufacturing" datasets?


* What do the authors mean by the second sentence in the Abstract
> This work has consisted of connecting encoders and decoders through attention mechanisms
or having multi-layer perceptrons.

And how is this related to the claims about "bigger is better" paradigm.

* Sec 4 is trivially split into subsections. Hampers readability

**Questions:**

See weaknesses above.

---

### Official Review · Reviewer_vPPQ · 2025-11-04

**Soundness:** 2
**Presentation:** 3
**Contribution:** 2
**Rating:** 4
**Confidence:** 4

**Summary:**

This paper provides a detailed theoretical introduction and experimental verification for selective embedding (SE). Both theory and experiment are comprehensive and thorough. Specifically, the manuscript claims that the method of data loading is crucial. By replacing the simple random shuffle strategy with strategies such as modality alternation and oversampling, the accuracy of the model can be improved. In particular, the manuscript provides several theoretical explanations for the effectiveness of the SE method, including enhancing diversity, increasing the effective sample size, and reducing the covariance between samples. Overall, although the method is not novel, the theory is detailed, the experiments are thorough, and the manuscript is well-organized and easy to follow.

**Strengths:**

- The manuscript is well-organized and is easy to follow and understand
- The theory is simple yet sound and comprehensive.
- The experimental results cover a wide range of fields including medicine and heavy industry, with a significant span.

**Weaknesses:**

- The issue that concerns me the most is that the work claims that the size of the dataset is not important, but this seems inconsistent with the content of the work. The focus of this work appears to be on improving the efficiency and accuracy of optimization through an optimized data loader. This does not seem to conflict with the size of the dataset itself. In other words, I think the work should prove that using the SE strategy can improve task accuracy and data efficiency, rather than reducing the size of the used dataset.
- Although the author may be concerned that the excessive introduction of the SE method might affect the innovation of the paper, I believe that the main focus of this work should be on theoretical explanations and extensive experimental verification. Therefore, the specific strategies of SE should be presented in detail. This aspect is notably lacking in the current manuscript; Chapter 3 is too short and it is difficult to follow.
- Considering that most ICLR readers may not be familiar with the fields of heavy industry and medicine, I suggest that the author provide more detailed information about the datasets. From the current perspective, at least the validation set for medical images is very small, consisting of only a few hundred images. This significantly reduces the credibility of the experimental results.
- I approve of the theoretical explanation of this work, although it is simple, it is comprehensive and sound. However, theoretical results should be verified through experiments. For instance, if the author claims that SE has improved the optimization efficiency, then a training curve should be drawn to demonstrate this.
- Although the current experimental results cover heavy industry and medicine. However, the result is singular, with only the precise result. The author should provide more validations based on the theoretical results. For instance, the changes in gradients (variances), individual ablations for SE method components (such as oversampling and data source alternation), and validations of generalization.
- Despite all the aforementioned flaws, I still appreciate the concise and effective work. I observed a similar approach in JMP (a molecular and materials-based model from Meta), so I am confident in the effectiveness of this work. However, from the perspective of the ICLR conference, I believe that this work is lacking in considerations of the main claimed points, the richness of the visual content, the verification of various theoretical results, and the completeness of the experimental setup. Even though it might be difficult, if the author can fully address the issues mentioned, I will increase my score.

**Questions:**

1. I need to see more validation of the theoretical results, such as the entropy, gradient covariance, and effective sample size mentioned in the text. Just having high accuracy is not enough for me to believe the multiple theoretical results presented in this paper, which is one of the most significant contributions of this paper.
2. The selection of the data set for the paper seems rather unreasonable. For instance, the validation set for medical images appears to be very small, and it seems that the test set has not been mentioned? In such a small dataset, if possible, cross-validation or increasing the size of the validation set should be adopted. Considering that the paper claims a significant improvement in accuracy, I believe it is crucial to illustrate this point in order to enhance the credibility of the experiment.
3. I suggest that the author provide a more detailed explanation of the methods used in the paper. The current Section 3 is too short. Good work should openly acknowledge the limitations and innovations of the research.
4. As a suggestion, the author seems to deliberately aim to illustrate the universality of the SE method. If so, I suggest that there should be at least three application scenarios, and they should not be scenarios like heavy industry which are relatively rare. Even so, I think the author should focus more on providing more thorough experimental evidence to support his theoretical conclusions.

---

### Note · Authors · 2025-11-20

I have read and agree with the venue's withdrawal policy on behalf of myself and my co-authors.